# Intergenerational Support and Second-Child Fertility Intention in the Chinese Sandwich Generation: The Parallel Mediation Model of Double Burnout

**DOI:** 10.3390/bs13030256

**Published:** 2023-03-14

**Authors:** Wenxiao Fu, Wenlong Zhao, Fei Deng

**Affiliations:** 1School of Humanities and Social Science, Xi’an Jiaotong University, Xi’an 710049, China; 2School of Education, Xi’an International Studies University, Xi’an 710128, China

**Keywords:** intergenerational support, parental burnout, burnout in caring for grandparents, fertility intention, sandwich generation

## Abstract

In 2013, the Chinese government implemented a two-child policy to increase the country’s fertility rate. However, the persistently low rates necessitated other measures to boost fertility. This study empirically investigated the association between intergenerational support and second-child fertility intention in the Chinese sandwich generation and demonstrated the mediating role of parental burnout and burnout in caring for grandparents. Survey data collected at Time 1, Time 2, and Time 3 included 2939 participants from different regions of China. Before analyzing the data, coarsened exact matching and propensity score matching was conducted to reduce sampling bias. Regression analysis results indicated that intergenerational support has a significant total positive effect on second-child fertility intention. Furthermore, mediation path analysis revealed that parental burnout and burnout in caring for grandparents play significant but opposite directional mediating roles in the association between intergenerational support and second-child fertility intention. Sensitivity analysis using different calipers yielded similar results. These results indicated that second-child fertility intention can be increased among the Chinese sandwich generation with intergenerational support, by mitigating parental burnout. However, intergenerational support did not alleviate burnout in caring for grandparents in the sandwich generation; therefore, formal older adult care policies are required to help the sandwich generation experience lower burnout, while receiving intergenerational support.

## 1. Introduction

China’s National Bureau of Statistics stated that the nation’s total fertility rate declined from 2.75 in 1980 to 1.15 in 2021 [1]. The 50 year-long one-child policy in China has not only restricted reproductive behavior and changed traditional fertility intentions, but has also created a unique Chinese sandwich generation. In China, the sandwich generation is characterized by only children, who bear heavier care burdens in parenting and caring for older adults than sandwich generations in other countries [2]. Under the threat of dual stress, the Chinese sandwich generation is at high risk of developing psychological problems, such as parental burnout (PB) and burnout in caring for grandparents (BCG), which may affect their intention to have more children [3]. In such circumstances, most Chinese grandparents will provide extra help to alleviate this double pressure.

Existing research has significantly explored fertility intentions in China; however, some important issues remain unexplored [4]. On the one hand, few studies have examined the combined effects of dual burnout on the fertility intentions of the sandwich generation. Given their shared impact on fertility intentions, the failure to combine these two types of burnouts may lead to misleading conclusions. On the other hand, many studies have confirmed the effect of intergenerational support (IS) on fertility intentions; however, little is known about the mechanisms underlying how IS affects second-child fertility intention (SCFI).

Considering these arguments, this study aimed to explore the mediating role of PB and BCG between IS and SCFI. In other words, this study sought to understand as to whether IS increases SCFI by mitigating the double burnout of the sandwich generation. These findings may provide a pathway to alleviate this double burnout, and accordingly offer insights into how best to address China’s low fertility rate. In addition, the findings could also be informative for other countries with very low fertility rates and fertility intentions (e.g., Republic of Korea).

## 2. Literature Review and Hypothesis Development 

### 2.1. Cooperative Breeding Hypothesis

The cooperative breeding hypothesis suggests that humans have developed a system of “cooperative breeding” in childrearing—specifically, mothers can get help from others to raise their children better [5]. The findings of the cooperative breeding hypothesis correspond with some earlier studies by anthropologists who found that women living in extended families were more inclined to have more children [6]. Sear and Coall reviewed the literature exploring family members helping mothers raise their children, and found that they cannot raise too many children on their own, and that some family members are important for child development and fertility [7]. In summary, we contend that the cooperative breeding hypothesis provides theoretical guidance for examining the impact of IS on the SCFI of the sandwich generation in China.

### 2.2. Intergenerational Solidarity Theory

The relationship between parents and children is special and complex; moreover, for a long time, no theory had directly explored intergenerational relationships. It was not until 1991 that Bengtson and Roberts proposed the intergenerational solidarity theory, which provided an important framework for exploring intergenerational relationships. The intergenerational solidarity theory is a multi-dimensional structure comprising six dimensions: affectual, associational, consensual, functional, normative, and structural solidarity [8]. A recent review mentioned that this theory provides a more comprehensive framework for investigating intergenerational support [9]. We contend that intergenerational solidarity theory may offer a suitable theoretical framework to test the underlying mechanism of IS and SCFI. 

### 2.3. Intergenerational Support and Second-Child Fertility Intention 

In 2016, the Chinese government implemented a two-child policy, which allows all married couples to have two children. Although the total fertility rate rose briefly to 1.6 in 2017, it declined to 1.15 by 2021. This reveals that the current family-planning policy has played only a limited role in increasing fertility rates. Globally, the number of countries entering aging societies is increasing, as is the proportion of the population with grandparents. Currently, grandparents make up 20% of the world’s population [10]. Although grandparents are not considered to be a productive population, they contribute to society by providing support to their children and increasing “family capital” [11]. Although the support provided by grandparents is voluntary and unpaid, we cannot ignore the positive effects of IS, such as a higher fertility rate.

In contemporary society, childbirth costs are rising rapidly, and having more children means requiring more resources from parents. Further, multiple pressures dilute the resources of the sandwich generation, driving couples to turn to their parents for additional resources. The existing literature indicates that the support provided by grandparents involves three main aspects. First, grandparents provide practical help, such as child care and household chores. According to China’s Research Center on Aging, 70% of children under age six are cared for by at least one grandparent [12]. Data from the U.S. and some European countries show that more than a quarter of families have grandparents to help with childcare and take care of some household tasks [13]. Gu et al. provided evidence from China that grandparents helping with childrearing can boost child fertility [14]. Secondly, grandparents also provide financial support, in order to reduce their children’s financial stress. Using data from 13,626 grandparents, Luo et al. showed that more than 60% of grandparents provided care for their grandchildren during a 10 year period, with higher-income grandparents providing additional financial support, such as paying for formal childcare costs [15]. Finally, researchers have confirmed that emotional assistance from grandparents affects parents’ attitudes and actions toward childbearing and parenting. Sear explored the link between support from grandparents and fertility, and confirmed that emotional support increases the likelihood of subsequent births [16]. Using longitudinal data from France and Norway, Tanskanen and Rotkirch suggested that both emotional support and child-care help from grandparents are associated with increased fertility intentions [17]. Therefore, we propose the following:

**H1.** *IS plays a positive role in SCFI*.

### 2.4. Intergenerational Support, Parental Burnout, and Second-Child Fertility Intention 

Although existing studies confirm the positive aggregate effect of IS on having more children, some unanswered questions still exist. On the one hand, the question of the underlying mechanisms between IS and fertility intention remains unclear, and on the other hand, existing studies have not explored IS and fertility among the sandwich generation in China. For this generation, the one-child policy has left them with the burden of raising children and also caring for grandparents. Whether IS can increase fertility by alleviating the double burden of the sandwich generation is the next question for further exploration.

Raising children is very demanding, especially for the sandwich generation. When parents are chronically unable to access additional resources to deal with parenting stresses, they face the threat of PB. PB is defined as “a state of intense exhaustion related to one’s parental role, in which one becomes emotionally detached from one’s children and doubtful of one’s capacity to be a good parent” [18]. Chen et al. showed that the average parental burnout score for Chinese parents was 48.03 (0–138), with a prevalence rate of 3% [19]. So far, researchers have focused on exploring the factors contributing to PB. Few researchers have explored and provided empirical evidence for factors that mitigate PB. According to the initial theoretical framework of PB (the balance between risks and resources (BR^2^)) theory, risk factors that can lead to high parenting stress levels are antecedents of PB, while adequate parenting resources can alleviate parenting burnout [20]. As an additional resource, IS may reduce parenting stress and mitigate PB. Thus, we propose:

**H2a.** *IS is negatively related to PB*.

A recent study of 17,409 parents from 42 countries demonstrated that the prevalence of PB varies globally, ranging from 1% to 10% [21]. The notion of PB has gained widespread attention, due to its high prevalence and serious consequences. Parents threatened by PB may lose their sense of fulfillment in raising children and begin to reject their parental role, eventually becoming increasingly emotionally distant from their children [22,23]. Yang et al. demonstrated that parental burnout was not only harmful to parents’ mental health but also predicted mental health problems (e.g., depression) in children [24]. A recent article shows that overwhelming parenting stress reduces a couple’s intention to have another child [25]. Given that parenting stress is one of the core features of PB, we infer that PB also reduces the likelihood that couples will have more children. Therefore, we propose the following: 

**H2b.** *PB is negatively related to SCFI*.

### 2.5. Intergenerational Support, Burnout in Caring for Grandparents, and Second-Child Fertility Intention

In China, adult children are not only influenced by Confucian culture, but are also governed by law to support their parents. Confucian culture considers adult children supporting their grandparents as an expression of filial piety, and China’s civil code includes legal obligations for children to support older adults, which has resulted in adult children and parents being embedded, by moral norms and laws, into a network of lifelong mutual obligations. Additionally, due to the one-child policy, many families lack enough adult children to share the burden of caring for parents. Thus, for the sandwich generation, a future of providing care for parents as they age is predictable. As a result, burnout from the stress of supporting older adults is likely to affect the fertility plans of the sandwich generation. Similar to PB, BCG can be defined as a syndrome that arises from supporting older adults in response to the stress caused by the caregiving environment. BCG also contains three dimensions: emotional exhaustion, depersonalization, and loss of accomplishment [26]. Although some researchers have suggested focusing on the effects of BCG in sandwich generation studies [27,28], few have done so. In summary, this study argues that there is a need to test the role played by BCG in the model. 

The existing literature does not agree on the relationship between IS and BCG. On the one hand, some researchers have argued through qualitative and quantitative analysis that IS can reduce caregiver burden and stress. Using data from 56 primary informal caregivers (mostly adult children or adult grandchildren), Reid et al. confirmed that the burden on caregivers would be significantly reduced if reciprocity existed between the care receiver and the caregiver [29]. In a recent article, Japanese researchers conducted open-ended interviews with 50 pairs of caregivers and care receivers, and confirmed that IS can reduce the burden of caring for parents by adult children [30]. On the other hand, some researchers have concluded, from an equitable reciprocity perspective, that grandparent support is not a purely altruistic act, but more of a filial investment. Offer further explained that there are three stages of reciprocity—giving, receiving, and repaying—and that the pressure to repay is most likely to be the main burden for people, which is also the main cause of psychological burden for the recipient [31]. As Blau asserts, reciprocity constitutes “a necessary condition of exchange that individuals, in the interest of continuing to receive needed services, discharge their obligations for having received them in the past” [32]. Non-reciprocated behaviors will usually lead to the breaking-off of social ties. Grandparental support is the giving component of reciprocity, and after receiving this support, adult children need to reciprocate to maintain intergenerational interactions, which may increase the burden on adult children and induce psychological burnout [33]. Therefore, we propose the following:

**H3a.** *IS is positively related to BCG*.

In China, adult children are the primary means of supporting older adults, but the long-term one-child policy has greatly reduced the birth rate, which has led to a serious reduction in the number of caregivers supporting older adults. Will adult children plan to have more children when supporting older adults becomes a major source of stress? Although the literature on BCG and fertility intention is limited, a few researchers have explored the relationship between older adult care-related stress and fertility. Exploring the support provided by adult children with full-time jobs in Canada to support their aging parents, Duxbury et al. noted that caring for older adults is strongly associated with absenteeism, lower well-being, and lower fertility, effects that are stronger for women [34]. The results of a recent quantitative article by Japanese researchers Sakata and McKenzie suggest that the stress of caring for parents directly reduces the number of children born to Japanese couples [35]. In addition, Sakata and McKenzie used the 1998 and 2008 National Family Research of Japan surveys to explore the link between older adult care-related stress and child quality, and confirmed that parental caregiving stress affects the quality of education for the third and subsequent children [36]. Considering that caregiving stress is an important indicator of BCG, we propose the following:

**H3b.** *BCG is negatively related to SCFI*.

### 2.6. Intergenerational Support, Parental Burnout, Burnout in Caring for Grandparents, and Second-Child Fertility Intention

Most previous studies on IS and fertility intentions have concluded that IS can increase fertility or fertility intentions. However, a few studies still conclude that IS can only increase fertility in specific populations or even that there is no direct relationship between IS and fertility [37]. An important reason for the inconsistent findings is the possibility of other factors influencing the path between IS and fertility intention—that is, whether IS affects fertility intention through other variables. IS reduces the financial and time pressures on couples, who have more resources to care for older adults and raise children and therefore have a higher likelihood of having more children. However, sandwich generations are aware of the need to provide care as their parents age in return for present-day support; therefore, they are likely to delay or cancel childbearing plans to avoid conflicts in allocating future support and resources. The exploration of the underlying mechanisms echoes the call of Sear and Coall. In a literature review, they noted that the relationship between IS and fertility intentions varies depending on the mediating variable [7]. We argue that IS may affect parents’ fertility intentions through PB and BCG. Moreover, researchers have pointed out that based on a significant total effect, if there is a significant mediating effect, then the direct effect of the independent variable on the dependent variable needs to be further tested to explain whether the indirect effect is fully or partially mediated [38]. Full mediation indicates that the researcher has fully revealed the mechanism of the independent variable’s influence on the dependent variable, and that there is no other mediating influence. Partial mediation implies that the independent variable’s influence on the dependent variable is other than the mediating factors in the model, implying the plausibility of other causal mechanisms. For the sandwich generation, it is possible that BCG and PB are mediating variables between IS and SCFI, but this does not determine the absence of other influencing mechanisms. For example, in a recent study, Wang and Zhao demonstrated that IS improves the odds of couples having a second child by alleviating incompatibility in maternal time allocation [39]. As expected, we propose the following:

**H4a.** *PB acts as a mediator between IS and SCFI*.

**H4b.** *BCG acts as a mediator between IS and SCFI*.

**H4c.** *There is a significant direct effect between IS and SCFI*.

### 2.7. Factors Influencing Intergenerational Support

IS is not random. Unobserved covariates are likely to exert effects on both IS and SCFI, confounding the causal effects between variables. Factors that may influence grandparents’ willingness to provide support, such as their own health status and their own, as well as their children’s, economic statuses, are crucial variables to control for when testing the effect of IS on fertility intentions [40]. Referring to the influential factors used to examine IS in previous studies, *hukou* (household registration in China), age, grandparents’ health and income, and other variables were covered in this study. Additionally, IS should consider variables related to both grandparents and adult children. Adult children with greater needs are more likely to expect support from grandparents than adult children with fewer needs. Therefore, adult children’s economic conditions and socioeconomic status are also included. Since different factors may influence IS and further affect fertility intention, particular methods are required, in order to reduce the selection bias, before analyzing the relationship between IS and SCFI. Figure 1 shows the study proposed model.

## 3. Methods

### 3.1. Participants and Data Collection

An ethics approval of the research was granted by the Human Research Ethics Committee (HREC), Xi’an JiaoTong University prior to data collection. We surveyed parents with one child in six Chinese cities in 2020 and 2021. The Center of Health Supervision National Commission of the PRC conducted a survey of more than 5.8 million couples who only had one child. We found that 95% of their first children were under 15. Therefore, the “parents of children currently in kindergarten, elementary school, and junior high school” were selected as respondents in this study.

To select respondents, the survey employed a multistage random sampling procedure based on levels of region, city, school, grade, and class. We selected six cities of different types (different economic development levels, population sizes, and social cultures) from different regions (eastern, central, and western) as a good representation of the different regions of China (Table 1). 

We extracted approximately 300 children, 600 primary school students, and 300 junior high school students from each city and county. Subsequently, the students’ parents (approximately 7232) were invited to participate in the survey.

Data were collected from respondents through online questionnaires, which were distributed by the head teachers via WeChat and QQ (the two most popular applications for information exchange in China). Web questionnaires have the advantages of low cost and convenient operation, but they also have certain limitations [41]. On the one hand, although the online questionnaire format allows respondents to fill out the questionnaire anytime and anywhere, the high degree of freedom in filling out the questionnaire and the limitation of the researcher not being able to be present in the room makes it impossible for the researcher to effectively control the processes of questionnaire distribution and filling out the questionnaire, which, in turn, does not guarantee an effective response. On the other hand, after the questionnaire link is sent to the online platform, the respondents may redistribute the questionnaire again. Therefore, in the questionnaire diffusion process, respondents may be both the respondent and the diffusers, which partly deprives the researcher of control over the questionnaire to some extent. Therefore, there is a need to technically control the forwarding privileges of the questionnaire when distributing the online questionnaire, in order to ensure the researcher’s control over the questionnaire.

Before the survey began, participants were asked to consent to participate. To reduce common method variance, this study was designed to conduct three measurements. The independent variables (IS) and demographic variables were measured at Time 1, while the mediators (PB and BCG) were measured at Time 2, with a four-month interval. Three months later, the outcome variable (SCFI) was measured at Time 3.

At Time 1, 7232 online questionnaires were distributed, and after excluding invalid questionnaires, 6901 were usable at a 95% response rate. Approximately four months later, we distributed a second questionnaire and received 6346 valid questionnaires with a response rate of 91%. At Time 3, we collected a total of 3812 questionnaires, with a return rate of 60%. Given that our study focused on the SCFI of sandwich families, we excluded those without grandparents and those that already had two or more children. The included sample consisted of 2939 parents. The survey respondents were from different socioeconomic and sociocultural backgrounds. Specifically, approximately 49.1% were men and 50.9% were women. Furthermore, 56% of the respondents had urban *hukou* (household registration system in China), and 44% had rural *hukou*. The majority of the respondents (95%) were between the ages of 20 and 49 years old, with a minority being over 50.

### 3.2. Measures 

This study uses scales validated in previous research to measure IS, PB, BCG, and SCFI. The scales were translated and back-translated with the help of three researchers in related fields, and minor revisions were made to make the scales applicable to Chinese culture. All scales can be found in the Appendix B (Table A1, Table A2, Table A3 and Table A4).

#### 3.2.1. Intergenerational Support

An adapted version of the Intergenerational Support Scale (ISS) was used in this study [42]. We asked parents whether they received support from their parents along three dimensions (i.e., financial, emotional, and practical). Each dimension included one question (“Do you receive financial/emotional/practical support from your parents?”). The following choice case was coded as either 1 (with respondents selecting “yes” to at least one dimension, representing the presence of IS) or 0 (representing the absence of IS). 

#### 3.2.2. Parental Burnout 

PB was assessed with the Chinese version of the parental burnout assessment (PBA) [43]. The scale consists of 23 items related to four subscales: exhaustion in one’s parental role (9 items), contrast with previous parental self (6 items), feelings of being fed up with one’s parental role (5 items), and emotional distancing from one’s children (3 items). PBA items were rated on the 7-point Likert scale—from *never* (0) to *every day* (6). The total score for all 23 items ranged from 0 to 138, with higher scores indicating higher levels of PBA. In the current sample, Cronbach’s alphas were 0.90, 0.88, 0.79, and 0.80 for the four subscales, and 0.92 for the global score. All the estimated factor loadings found in the CFA were significant at *p* < 0.001. Standardized factor loadings ranged between 0.79 and 0.91. In the fit indices, CFI = 0.96, TLI = 0.95, RMSEA = 0.04, and SRMR = 0.03. These results confirm the reliability of the PBA scale and the validity of its four-factor internal structure. Results are detailed in Table 2.

#### 3.2.3. Burnout in Caring for Grandparents 

An adapted version of the Maslach Burnout Scale—Human Services Survey (MBI-HSS) was used to measure BCG [44]. The scale contains 22 items and is divided into three subscales: emotional exhaustion (9 items), depersonalization (5 items), and personal accomplishment (8 items). The MBI-HSS is a 7-point Likert scale, with answers ranging from 0 (*never*) to 6 (*every day*). As such, a high score means a high level of BCG. In this study, the internal consistency was good to excellent for the three subscales, and the Cronbach’s alphas were 0.92, 0.73, 0.85, and 0.88 for the global score. All the estimated factor loadings found in the CFA were significant at *p* < 0.001. Standardized factor loadings ranged between 0.76 and 0.93. In the fit indices, CFI = 0.92, TLI = 0.93, RMSEA = 0.05, and SRMR = 0.04. These results confirm the reliability of the BCG scale and the validity of its three-factor internal structure. Results are detailed in Table 2.

#### 3.2.4. Second-Child Fertility Intention

This study uses an adapted version of the Fertility Intention Scale to collect data on the intention to have a second child [45]. The scale contains 5 items (e.g., Do you intend to have a second child in the following years?), and each item was scored using a 5-point Likert scale, with higher scale scores meaning greater second child fertility intention. In this study, Cronbach’s alpha of the scale was 0.89. All the estimated factor loadings found in the CFA were significant at *p* < 0.001. The standardized factor loading is 0.95. In the fit indices, CFI = 0.96, TLI = 0.94, RMSEA = 0.03, and SRMR = 0.02. These results confirm the reliability of the SCFI scale and the structure validity. Results are detailed in Table 2.

#### 3.2.5. Control Variables

This study used a set of control variables to estimate the propensity scores, specifically, paternal grandparents’ *hukou*, socioeconomic status, and health conditions, as well as the fathers’ age, occupational reputation, the first child’s gender, maximum years of education, whether subjects were living with parents or not, and the age of the first-born child. The *hukou* system was measured as a dichotomous dummy variable, coded as 1 for rural and 2 for urban. This study built the International Socio-Economic Index of Occupational Status (ISEI) based on the International Standard Classification of Occupation (ISCO-88) and used Treiman’s Standard International Occupational Prestige Scale (Treiman’s SIOPS) to measure participants’ occupational reputation. This scale measures grandparents’ health conditions, with the answers coded as 1 = “*very bad*” to 5 = “*very good*”. 

#### 3.2.6. Analytical Strategy 

Descriptive analysis, Chi-squared tests, and *t*-tests were first conducted on the general characteristic of variables. To address the selection bias, coarsened exact matching (CEM) for coarsening pruning was first performed by Stata 16.0, and then the nearest neighbor matching with put-back was performed by the PSMATCH2 program in the Stata 16.0 package. Subsequently, the matching effect was further tested for balance and sensitivity.

Then, regression and mediation analyses were performed using the post-matching sample. Since both the mediating and dependent variables are continuous variables, this paper uses ordinary least squares (OLS) regression to analyze the relationship between the variables. We built an SEM model, following the approach introduced by Preacher and Hayes, to test the multiple mediation effects [46]. Moreover, this paper uses bootstrapping to calculate the mediation effect, mainly because this method does not require the sample to be normally distributed, which greatly reduces the type I error rate and obtains valid asymptotic refinement estimates. Mplus is preferred over other applications because it provides not only the total indirect effects, but also each specific indirect effect, so that comparisons between different indirect effects can be easily made. Therefore, all the mediation analyses were conducted in Mplus 8.0.

Before matching, all variables had missing data rates of around or below 1%. We used multiple imputations to impute missing data. After matching, all variables within the matched sample had no missing values. Additionally, to test the validity and robustness of the assumption that unmeasured covariates are ignorable, the balancing test and Rosenbaum sensitivity test procedures were run.

## 4. Results

### 4.1. Descriptive Statistics, Chi-Squared Tests, and t-Tests before Matching

This study first conducted descriptive statistics on the variables to determine the differences between families with and without IR. As shown in Table 3, 67.98% of the families had IS, while 32.02% did not. The families with IS reported a higher intention to have a second child. In addition, PB was slightly lower in families with IS. In terms of BCG, the score of the two groups was equal. For the control variables, there were significant differences between the two groups. Detailed results are shown in Table 3. 

### 4.2. Results of CEM and PSM

King and Nielsen note that “the more balanced the data, or the more balanced it becomes by pruning some observations through matching, the more likely PSM will degrade inferences—a problem we refer to as the PSM paradox” [47]. Therefore, the first step in using PSM is to detect the balance of the original data. Based on the *t*-test in Table 1, there is indeed a significant difference between the two groups, indicating that the raw data would be more balanced using PSM. However, PSM also suffers from inefficiencies, model dependencies, and excessive data pruning. King and Nielsen suggest that if researchers need to use PSM, they can first use CEM for coarse pruning, before using PSM for exact matching after carefully checking for imbalances. Recently, it has also been pointed out that there is no single matching method that can reduce the bias across all scenarios [48]. Therefore, in this study, based on the recommendation of King and Guo, it was decided to first use CEM for coarsening pruning and then PSM for nearest neighbor matching with put-back. After coarsening pruning with CEM, the total number of participants was 2228, including 1406 in the intervention group and 822 in the control group, and the overall imbalance *L*_1_ = 0.518. The *L*_1_ value itself has no value, but serves as a point of comparison between matching solutions. *L*_1_ = 0 indicates perfect global balance, while larger values indicate large imbalances between groups, with the maximum value *L*_1_ = 1, indicating complete separation. 

The estimation of the propensity value consists of two steps, the first being the selection of appropriate control variables and the second being the selection of the model to perform the estimation. There are various strategies for selecting variables, and this study chose the approach proposed by Heckman et al. [49]. This approach relies on statistical significance. Specifically, if a variable is significant at conventional levels, it will be retained. Notably, any discrete choice model can be selected, in principle. Due to the well-known shortcomings of the linear probability model, researchers prefer logit or probit models. In this paper, we used the logit model to estimate the set of variables. The variables and logit results are detailed in Appendix A. After obtaining the propensity values, this study was matched using the put-back nearest neighbor method, and the total number of respondents after CEM and PSM matching was 1445, including 827 in the intervention group and 618 in the control group.

### 4.3. Results of the Balance Test and Common Support

The post-PSM Chi-squared tests and *t*-test revealed that the standardized deviations of the variables mentioned above are significantly reduced, and the means of all the variables are not significantly different, indicating a better matching effect. According to Rubin, the results for the matched sample showed a B-value of 10.5, which is less than 25 and, therefore, acceptable, as well as an R-value of 0.71, which was within the acceptable level of 0.05 to 2 [50]. The results showed that PSM significantly reduced the difference between the two groups, minimized the sample selection bias, and satisfied the balancing property, which also means that unobserved covariates did not influence the test results. Matching results are detailed in Appendix A.

Besides, checking the overlap and the region of common support between treatment and comparison groups is important in determining the validity of PSM. In this study, there were adequately matched samples between the treatment and comparison groups and very few lost samples, suggesting that the use of this sample for data estimation was representative. The histogram of the common support region is detailed in Appendix A.

The new matched sample generated by PSM (N*_families with IS_* = 827; N*_families without IS_* = 618) was used for regression and mediation analysis. Table 4 indicated the associations between IS, PB, BCG, and SCFI. IS had a significantly negative effect on PB (β = −1.814, *p* < 0.05), indicating that families with IS report less PB. H2a was supported. In addition, there were positive and significant links between IS and BCG (β = 2.043, *p* < 0.05). Therefore, H3a was supported. To test H1, H2b, and H3b, we examined the direct effect of IS, PB, and BCG on SCFI. Model 3 shows that IS had a positive total effect on SCFI (β = 0.446, *p* < 0.01), which suggested that families with IS reported a higher SCFI. Both PB and BCG had a negative effect on SCFI (β_PB_ = −0.022, *p* < 0.05; β_BCG_ = −0.005, *p* < 0.05), which means the greater the PB and BCG, the lower the SCFI. Thereby, H1, H2b, and H3b were supported. 

Furthermore, a fitting structural equation model in Mplus and the bootstrap approach were used to analyze the mediation relationships between variables. In general, the indirect effects of IS on SCFI through two mediators were significant, with relevant variables under control. H4a and H4b were supported. More specifically, according to the results of 2000 iterations of bootstrap resampling, Table 5 presents the lower-level and upper-level confidence intervals that do not contain zero, which confirmed the indirect effect of IS on SCFI via PB, with BCG under control. Similarly, following the steps mentioned above, the lower-level and upper-level confidence intervals do not contain zero, confirming the direct effect of IS on SCFI through BCG, with PB under control. Comparing the two specific mediating effects, the difference is found to be 0.03, and the lower-level and upper-level confidence intervals do not contain zero, indicating that the mediation effect of PB was stronger than that of BCG. Additionally, there was a significant direct effect between IS and SCFI, i.e., PB, as well as BCG, were shown to play a partial mediating effect. H4c was supported. The *p*-values for all results were significant, and the detailed results are shown in Table 5.

### 4.4. Sensitivity Analysis

Matching effects are based on the conditional independence assumption. Hidden bias may occur if unobserved variables affect both the dependent and independent variables simultaneously. This problem can be fixed by adding additional variables, collecting additional data, or designing the experiment as a completely randomized experiment, but these strategies have challenges, such as being too costly and difficult to implement. A preferable approach would be to first perform a sensitivity analysis, in order to estimate the bias level. Two methods were used for sensitivity analysis in this study. On the one hand, different calipers were selected for matching to test the robustness of the matching results in this paper. On the other hand, the bounding approach proposed by Rosenbaum was used to detect hidden bias. In general, if the value of Γ is too small (Γ < 2), the model is too sensitive, and the problem of unobserved variables in the model is more serious. This study uses the Wilcoxon Signrank Tests, which are currently supported by most analytic software, in order to test the sensitivity of the matching analysis results. The result of the Wilcoxon Signrank Tests showed that the value of Γ was larger than 2.9, which satisfied the condition that Γ was greater than two, indicating that the model was more robust. Appendix A present the results of the sensitivity analysis. 

## 5. Discussion and Conclusions 

Using the post-PSM data, we aimed to develop a parallel mediation model to test the relationship between IS and SCFI, as well as to test the mediating role played by PB and BCG among the Chinese sandwich generation. The findings confirm that IS has a positive effect on SCFI and that it affects SCFI through dual burnouts; however, the impacts of the two mediators do not occur in the same direction. We proposed four sets of hypotheses to investigate the relationship between IS, PB, BCG, and SCFI.

The first set concerned the total effects of IS and SCFI. Before conducting the mechanism analysis, a total effects analysis of the independent and dependent variables was needed to prepare for calculating the mediating effects’ share. Regression analysis indicated that IS had a significant positive effect on SCFI. This finding was consistent with Harknett et al., who demonstrated that the IS environment was more strongly related to higher-order births than to first births [51]. Further, an article using data on Chinese working women confirmed the positive effect of IS on fertility [14]. However, the existing literature does not explore the mechanism of action between IS and fertility among the Chinese sandwich generation.

The second set of hypotheses concerns the mediation effects of PB on the link between IS and SCFI. The results showed that IR had a significant indirect effect on SCFI through PB, which meant that IS mitigates the PB of the sandwich generation, thereby increasing SCFI. Parenting is a very demanding task, and parents would have more resources to raise their children if they had access, such as grandparent support. Thus, one possible explanation for IS’s ability to increase SCFI by reducing parental burnout is that parents receive more support from grandparents, which reduces parenting stress and burnout, and ultimately leads to an intention to have more children.

The third set tested the mediating role of BCG. Regression analysis indicated that IS had a significant positive effect on BCG, which meant the more assistance the sandwich generation had with their parents, the higher their BCG level. The results of the structural equation in Mplus showed that IS had a significant negative indirect effect on SCFI through BCG. One possible explanation was that the greater the grandparents’ investment in the “filial bank”, the more care the sandwich generation needed to return to their parents, which increased the burden on adult children as caregivers. The study’s findings were similar to Offer’s [31]. Ultimately, IS reduced the fertility intentions of the sandwich generation through BCG. Moreover, comparing the effect sizes of the two parallel mediators, PB played a greater mediating role, which implied that IS increased SCFI by lowering the PB of the sandwich generation.

The fourth hypothesis tested whether the direct effect of IS on SCFI was significant. The results showed a significant direct effect of IS on SCFI, which also indicated that the mediating effect in the model was partially, rather than fully, mediated. Moreover, the results of partial mediation revealed that PB and BCG were only partially mediating causes in the pathways of IS and SCFI effects, and the possibility of other mediating mechanisms still existed. It is noteworthy that although the two mediating variables do not have the same direction of effect, both mediating effects were smaller than the direct effect. One possible explanation was that IS still increased or decreased SCFI through other variables, but, overall, the direct positive effect of IS on the sandwich generation was significant, which may imply that although the pathway of the positive effect of IS is complex, the intention of the sandwich generation to have a second child will increase if the frequency and support level are increased.

## 6. Implications

This study carries several theoretical implications. First, it proposed a dual mediator model to explore the relationship between IS and sandwich generation’s SCFI. Although existing articles confirm a positive relationship between IS and having more children, no research has revealed the underlying mechanisms between the two variables. This study confirmed the mediating effect played by PB and BCG and also revealed that PB and BCG play a partial mediating role, revealing the possibility of other mechanisms of action and providing a possible direction for further research. 

Second, as a concept that has only been clearly defined by researchers in recent years, PB and BCG have quickly sparked the attention of other researchers. Researchers have found that PB and BCG are more difficult to avoid and more prevalent than job burnout; accordingly, they deserve further exploration [52]. Although some Chinese scholars have keenly captured this cutting-edge research direction, few empirical articles have examined PB and BCG in Chinese parents, and even fewer have explored PB and BCG in the sandwich generation in the context of Chinese culture. Therefore, this study expanded the application of PB and BCG in the Chinese cultural context. 

Finally, the results of this study have implications for other countries with low fertility rates. Vollset et al. published an article on global fertility in *The Lancet*, noting the steady decline in global fertility and the continuing decline in fertility intentions [53]. Other countries that have had birth control policies (e.g., Iran, Vietnam) and very low fertility rates (e.g., Japan, South Korea) have a large sandwich generation that faces the same PB and BCG as the sandwich generation in China. Arguably, there is a global threat of lower fertility intentions. Therefore, the results of this study not only help to understand the double burnout faced by the sandwich generation in China and ways to enhance fertility intentions, but they also help to provide experiences and insights for other countries with similar social problems. 

This study also has important practical implications for demographers and policymakers, especially those concerned with the fertility rate. On the one hand, by empirically testing the main factors that can boost fertility intentions, we find that IS can significantly increase SCFI for the sandwich generation. To ensure more frequent intergenerational exchanges and intergenerational support, the Chinese government released a guide to create a better life for the elderly in November 2021, which promotes young adult children to live close to or co-reside with their parents. At the same time, the government will appropriately reduce personal income tax for adult children living with their parents, in order to more effectively protect the shared welfare of young adult children and the elderly. 

On the other hand, perhaps the most important finding of this study was that IS failed to reduce BCG in the sandwich generation. Therefore, more support is needed to help the sandwich generation care for older adults. Although the Chinese government advocates for the construction of formal care institutions for older adults, most older adults still reject them. Those children who send older adults to institutions are considered unfilial; hence, implementing formal institutions is very difficult. Du suggests that community-based services can be developed to meet older adults’ expectations to stay in their homes, while reducing the stress of the sandwich generation [54]. In short, while living in an environment filled with stress and burnout, the sandwich generation is likely to be reluctant to have more children. Therefore, building an environment in which parenting and caregiving are easier is most important for raising the fertility rate.

## 7. Limitations and Suggestions for Future Research

This study had several limitations. First, although the CEM and PSM approach could avoid self-selection bias in the independent variable, such bias in the mediating variables was still not properly addressed [55]. Thus, future research could address this issue, from support status to SCFI, by controlling for variables that may influence IS, PB, and BCG. Second, this study measured IS in three main dimensions, based on intergenerational solidarity; however, it was still insufficient to include all IS behaviors. Therefore, future research could measure IS from a broader perspective. Third, the parallel mediation model constructed in this paper is only a first step in exploring the mechanisms underlying IS and SCFI and does not discuss the relationship between the two mediators. Whether there is a competitive relationship between PB and BCG and whether there is a chain relationship are next steps for future research. Finally, although intentions may be a powerful predictor of fertility, they do not always match outcomes. Therefore, future studies should examine fertility trajectories instead of intentions. Specifically, future research could conduct a longer-term longitudinal survey, splitting the outcomes of SCFI into groups (e.g., intention—stability, intention—revision, intention—realization) and separately testing the relationship between the independent variable and the different SCFI outcomes.

## Figures and Tables

**Figure 1 behavsci-13-00256-f001:**
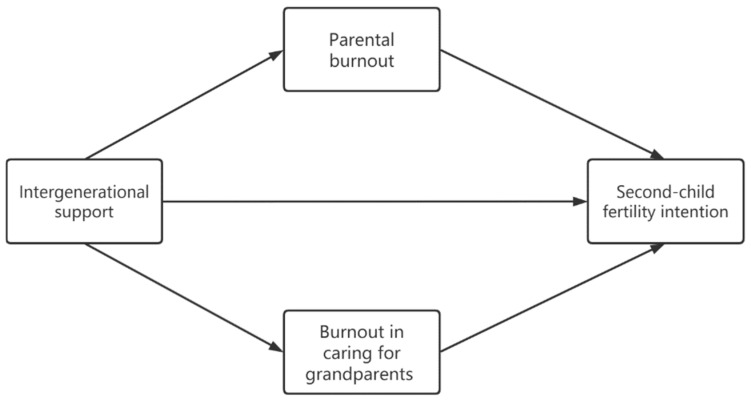
Proposed Model.

**Table 1 behavsci-13-00256-t001:** Surveyed Cities.

	Eastern Region	Central Region	Western Region
Cities	Fujian Xiamen	Hunan Changsha	Shaanxi Xi’an
Counties	Zhuhai Jinwan	Shanxi Yuci	Shaanxi Sanyuan

**Table 2 behavsci-13-00256-t002:** The results of CFA and its correlations.

		Items	Cronbach’s Alphas	std. Factor Loading	χ^2^/df	CFI	TLI	RMSEA	SRMR
PB	PB scale	23	0.92	-	5895.69 ***	0.96	0.95	0.04	0.03
	Subscale 1	9	0.90	0.91					
	Subscale 2	6	0.88	0.79					
	Subscale 3	5	0.79	0.85					
	Subscale 4	3	0.80	0.82					
BCG	BCG scale	22	0.88	-	4859.44 ***	0.92	0.93	0.05	0.04
	Subscale 1	9	0.92	0.93					
	Subscale 2	5	0.73	0.76					
	Subscale 3	8	0.85	0.88					
SCFI	SCFI scale	5	0.89	0.95	2140.70 ***	0.96	0.94	0.03	0.02
	IS			PB			BCG		
PB	−0.31 ***								
BCG	0.40 ***			0.23 *					
SCFI	0.39 **			−0.46 ***			−0.38 **		

Note. * *p* < 0.05, ** *p* < 0.01, *** *p* < 0.001.

**Table 3 behavsci-13-00256-t003:** Study sample characteristics, stratified by support status, before CEM and PSM.

	Total Sample		Intergenerational Non-Support	Intergenerational Support	
	Mean	SD	Mean	Mean	*p*
Independent variable					
Intergenerational (non)support	0.680	0.467			
Dependent variable					
SCFI	3.3667	1.010	2.879	3.596	<0.001 ^a^
Mediator					
PB	49.131	13.386	51.487	48.021	<0.001 ^a^
BCG	46.868	12.236	47.534	46.554	0.039 ^a^
Control variables					
Father’s socioeconomic status	2.416	0.893	2.239	2.499	<0.001 ^a^
Mather’s socioeconomic status	2.323	0.847	2.271	2.348	<0.001 ^a^
Father’s age	37.096	6.029	37.738	36.794	<0.001 ^a^
Mather’s age	36.266	6.028	36.021	36.381	<0.001 ^a^
Paternal grandfather’s socioeconomic status	33.448	11.995	32.812	33.748	<0.001 ^a^
Maternal grandfather’s socioeconomic status	31.874	10.254	31.354	32.119	<0.001 ^a^
Paternal grandparents’ health condition	3.808	0.995	3.411	3.995	<0.001 ^a^
Maternal grandparents’ health condition	3.749	1.024	3.486	3.873	<0.001 ^a^
Paternal grandparents’ *hukou* (1 = rural)	1.188	0.391	1.224	1.171	<0.001 ^b^
Father’s occupational reputation	40.339	8.715	39.473	40.747	<0.001 ^a^
Mather’s occupational reputation	39.636	8.716	38.184	40.320	<0.001 ^a^
First child’s gender (1 = men)	1.469	0.499	1.467	1.470	<0.001 ^b^
Maximum years of parents’ education	12.384	3.017	12.397	12.379	0.591 ^a^
Living with grandparents or not (1 = no)	1.239	0.425	1.107	1.301	<0.001 ^b^
Firstborn children’s age	8.521	5.261	7.165	9.160	<0.001 ^a^
N	2939		941	1998	

Note: ^a^
*p* values are from the t-tests(two-tailed), ^b^
*p* values are from the chi-square tests.

**Table 4 behavsci-13-00256-t004:** Results of the regression analysis after PSM.

Variables	PB (M1)		BCG (M2)		SCFI (M3)	
	Coefficient	SE	Coefficient	SE	Coefficient	SE
Father’s socioeconomic status	−1.271 *	0.561	−1.612	0.414	0.183 *	0.033
Mather’s socioeconomic status	−1.511 *	0.561	−1.609	0.414	0.192 *	0.033
Father’s occupational reputation	0.001	0.074	0.065	0.061	0.011 *	0.004
Mather’s occupational reputation	0.001	0.074	0.065	0.060	0.008 *	0.004
Father’s age	−0.038	0.072	0.084	0.064	0.005	0.004
Mather’s age	−0.041	0.072	0.092	0.064	0.006	0.004
Paternal grandfather’s socioeconomic status	0.002	0.062	−0.053	0.054	−0.003	0.003
Maternal grandfather’s socioeconomic status	0.002	0.012	0.021	0.043	0.004	0.002
Paternal grandparents’ health condition	−2.87 *	0.491	−1.314 *	0.427	0.317 **	0.029
Maternal grandparents’ health condition	−0.471	0.427	−1.272 *	0.383	−0.029	0.026
Paternal grandfather’s *hukou* (1 = rural)	−0.116 *	0.120	−1.523	1.065	−0.221 *	0.071
First child’s gender (1 = men)	0.047 *	0.019	0.128	0.094	0.227 *	0.147
Living with grandparents or not (1 = no)	−0.138	0.057	0.124	0.047	0.179 *	0.095
Firstborn children’s age	0.004	0.002	0.003	0.001	0.069	0.031
IS	−1.814 *	0.911	2.043 *	0.847	0.446 **	0.055
PB					−0.022 *	0.013
BCG					−0.005 *	0.002
Constant	61.286	4.410	51.731	3.917	1.727	0.312
R^2^	0.059		0.058		0.270	
F	5.41		5.38		28.39	

Note. * *p* < 0.05, ** *p* < 0.01.

**Table 5 behavsci-13-00256-t005:** Results of mediation analysis after PSM.

			Bootstrap BC 95% CI		
Effect	Estimate	SE	Lower	Upper	*p*-Value
IS->PB (a_1_)	−1.814	0.817	−2.322	−0.384	0.042
IS->BCG (a_2_)	2.043	0.847	0.741	3.544	0.009
PB->SCFI (b_1_)	−0.022	0.011	−0.020	−0.009	0.000
BCG->SCFI (b_2_)	−0.005	0.003	−0.009	−0.004	0.011
IS->SCFI (c’)	0.396	0.012	0.886	2.569	0.000
IS->PB->SCFI (a_1_*b_1_)	0.0404	0.014	0.005	0.038	0.040
IS->BCG->SCFI (a_2_*b_2_)	−0.010	0.005	−0.012	−0.005	0.050
a_1_*b_1_-a_2_*b_2_	0.03	0.015	0.009	0.053	0.008
Model fit information	CMIN/DF = 1.570; GFI = 0.945; TLI = 0.956; RMSEA = 0.071; SRMR = 0.080				

## Data Availability

The datasets generated during and/or analyzed during the current study are available from the author on reasonable request.

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
