# Peer review of "Intergenerational Support and Second-Child Fertility Intention in the Chinese Sandwich Generation: The Parallel Mediation Model of Double Burnout"

_behavsci, 2023, doi:10.3390/bs13030256_

Round 1

Reviewer 1 Report

The manuscript represents full potential to be published subject to minor revisions.  According to what the author(s) reviewed in the LR section, hypothesis 3a (H3a) should demonstrate a positive relationship. As a result, other parts should be revised accordingly pertaining to this hypothesis.

Author Response

A response to the comments made by the reviewer #1

We really appreciate you for your valuable comments and suggestions. According to the comments, we have revised our manuscript and the comments are answered one by one as follows. The reviewer’s comments are in italics, the authors’ responses are in red.

The manuscript represents full potential to be published subject to minor revisions.  According to what the author(s) reviewed in the LR section, hypothesis 3a (H3a) should demonstrate a positive relationship. As a result, other parts should be revised accordingly pertaining to this hypothesis.

Response: We thank the reviewer for raising the concern of this issue. Based on the literature review section, we have revised hypothesis 3a to “IS is positively related to BCG”. In addition, we have also revised other sections to ensure that the data analysis, conclusions, and suggestions sections argue for this hypothesis. In the revised manuscript, I have marked the changes regarding this issue in orange.

Reviewer 2 Report

Dear authors
My personal opinion is that after reading the paper, the manuscript is of potential interest to the readership of this journal, but there are issues that must be addressed:
Introduction
The positioning of the paper is not entirely clear. The author is better to explain the gap in this article further.
A concise introduction to enable the reader's understanding of the research problem.
what research gap the paper aims to fill, what contribution the paper provides, and why the contribution is important.
Literature review
The paper should relate coherently and convincingly with issues of real-world significance. This is a crucial phase contributing to research design.
Suggestions
• Add more information to enable readers' understanding of the authors' view.
Findings and discussion
Needs clear and comprehensive explanations to assist readers' understanding.

Reference.
- Using the following reference could be beneficial as these add more evidence to the literature review section:

 Investigating social capital, trust and commitment in family business: Case of media firms. Journal of Family Business Management, 12(4), 938-958.

 Best of luck with the further development of the paper.

Author Response

We really appreciate the Reviewer 2 for the valuable comments and suggestions. These comments are all very valuable and helpful for revising and improving our paper, as well as the important guiding significance to our researchers. We have studied comments carefully and have made correction which we hope meet with approval. The comments from Reviewer 2 and the corresponding revisions are attached in the pdf file.

Reviewer 3 Report

This is a well written and organized paper. I have some minor concerns.

The authors distributed two questionnaires at Time 1 and Time 2. It is unclear if the questions were the same or the two questionnaires were different, including two different sets of questions. Please, make this issue clear.

Measurement scales used in the study. As the study used scales validated in previous research to measure IS, PB, BCG, and SCFI, the authors should provide sources for the individual scales and include scales in an Appendix.

Generally, the SEM approach uses latent variables to account for measurement errors. As I understand, the authors did not used latent variables but only manifest variables. To what extent does such modeling approach provide a robust analysis?

Are unmeasured covariates which have been ignored related to interaction effects among variables?

Please, provide correlations between variables.

Author Response

We really appreciate the Reviewer 3 for the valuable comments and suggestions.

These comments are all very valuable and helpful for revising and improving our paper, as well as the important guiding significance to our researchers. We have studied comments carefully and have made correction which we hope meet with approval.

 The comments from Reviewer 3 and the corresponding revisions are attached in the pdf file.

Round 2

Reviewer 2 Report

Dear author(s)

Hope you are doing well. According to the review of this article, the corrections have been made.

Good luck